

# Effects of coupled hydro-mechanical model considering two-phase fluid flow on potential for shallow landslides: a case study in Halmidang Mountain, Yongin, South Korea

Sinhang Kang [1], Byungmin Kim [1]

[1]Ulsan National Institute of Science and Technology, 50, UNIST-gil, Eonyang-eup, Ulju-gun, Ulsan, Republic of Korea 44919

*Correspondence to*: Byungmin Kim (byungmin.kim@unist.ac.kr)

**Abstract.** More than 30 shallow landslides were caused by heavy rainfall that occurred on July 26 and 27, 2011, in Halmidang Mountain, Yongin-si, Gyeonggi Province, South Korea. To precisely analyze shallow landslides and to reflect the mechanism of fluid flow in void spaces of soils, we apply a fully coupled hydro-mechanical model considering two-phase fluid flow of

water and air. The available GIS-based topographic data, geotechnical and hydrological properties, and historical rainfall data are used for infiltration and slope stability analyses. Changes in pore air and water pressures and saturations of air and water are obtained from the infiltration analysis, which were used to calculate the safety factor for slope stability assessment. By comparing the results from numerical models by applying a single-phase flow model and a fully coupled model, we investigate the effects of air flow and variations in hydraulic conductivity affected by stress–strain behavior of soil on slope stability. Our

results suggest that air flow and hydro-mechanical coupling affects the rate of increase in pore water pressure, thus influencing the safety factor on slopes when ponding is more likely to occur during heavy rainfall. Finally, we conduct slope failure assessments using the fully coupled model, slightly more consistent with actual landslide events than the single-phase flow model.

## 1 Introduction

Rainfall-induced landslides can contribute to serious human casualties and cause property damage (e.g., Kim et al., 2017; Kang and Lee, 2018). Several factors such as the geotechnical and hydrological properties, geological characteristics, topographic features, and slope surface cover affect initiation of rainfall-induced landslides (Cho, 2014). Heavy rainfall can cause slope failures because of the infiltration of rainfall into the slopes, triggering the following sequential changes: an increase in the unit weight of soil because of an increase in water content (Ng and Shi, 1998; Cho and Lee, 2001); a decrease in the effective

stress owing to the loss of matric suction (Yeh et al., 2008); and a decrease in the shear strength.

Empirical methods have been frequently utilized to assess the potential of shallow landslides at large areas, but deterministic physics-based methods are preferable to accurately understand the mechanisms of landslides (Thomas et al., 2018). Numerous





studies have been conducted depending on physically-based methods to investigate the rainfall-infiltration behavior. Several studies have maintained assumptions to simplify solving the differential equation of Richards (1931) using a single-phase fluid

flow model of water without considering air flow (e.g., Ng and Shi, 1998; Cho, 2014; Tran et al., 2017). Notably, the ground comprises three phases of material (i.e., solid (soil particles), water, and air), and air flow caused by the interaction of water and air during infiltration greatly affects the infiltration behavior (e.g., Touma and Vauclin, 1986; Sun et al., 2015; Dong et al., 2017; Wu et al., 2017). Therefore, the effect of air flow on rainfall-infiltration has to be considered. Cho and Lee (2001), Borja and White (2010), and Borja et al. (2012) applied air pressure in the void spaces of a partially saturated soil for rainfall-

infiltration analyses, but it was assumed that the air pressure is equivalent to the atmospheric pressure. A two-phase fluid flow of water and air has been applied to more accurately investigate rainfall-infiltration (e.g., Zhang et al., 2009; Sun et al., 2015; Cho, 2016; Hu et al., 2018; Liu et al., 2018). Soil structure varies during infiltration of water because of changes in porosity which is sensitive to the level of saturation and the stress modification caused by infiltration-induced deformation (Wu and Selvadurai, 2016), contributing to changes in permeability (Hu et al., 2011). In this study, therefore, we applied a coupled

hydro-mechanical model considering two-phase fluid flow based on numerical methods to evaluate the slope stability using an infinite slope in a two-dimensional domain.

Previous studies have assessed slope stability on regional and local scales to simulate actual landslide events. Simoni et al. (2008) applied GEOtop-FS (Rigon et al., 2006) to simulate shallow landslides, in which the subsurface flow was modeled based on the numerically integrated 3-D Richards equation and on a given topography. However, the simulation in a 3-D

scheme requires a substantial amount of computation. Park et al. (2016), Tran et al. (2017), and Tran et al. (2018) have evaluated slope stabilities on regional and local scales using TRIGRS (Baum et al., 2008), TiVaSS (An et al., 2016), and Scoops3D (Reid et al., 2015), respectively, but both air flow and hydro-mechanical coupling were not considered. Coupled hydro-mechanical modelling considering two-phase flow needs to be applied to evaluate slope stability on a regional scale to suitably model slope failures with efficient uses of computing resources.

We simulated rainfall-infiltration in a 2-D scheme by applying both the single-phase flow model and the coupled hydro-mechanical model considering two-phase flow. The changes in pore air/water pressures and void ratios obtained from the simulation of rainfall-infiltration were used as input data for slope failure analyses at each cell of the GIS-based topography of Halmidang Mountain located in Yongin-si, South Korea. The performance of the slope failure assessments using the coupled hydro-mechanical model was then evaluated. In this study, we (1) evaluate the effects of air pressure and flow on the infiltration

behavior by comparing changes in pore water pressure and matric suction simulated by a single-phase flow model with those simulated by the coupled hydro-mechanical model, (2) investigate the effect of air pressure and flow on safety factors during localized heavy rain, (3) compare the landslide susceptibility map that was produced by application of the single-phase flow model and that produced by application of the coupled hydro-mechanical model, and (4) evaluate the applicability of the





single-phase flow model and the coupled hydro-mechanical model by a comparison of the areas predicted as unstable with the
initiation location of actual shallow landslides.

## 2 Study area

Halmidang Mountain is located in Yongin-si, Gyeonggi Province, South Korea, at 37°18′30″–37°21′50″N and 127°10′10″–
127°14′70″E, and it covers approximately 15 km². Figure 1(a) shows the elevation distribution of the Halmidang Mountain
based on 1:5000 digital maps provided by the National Geographic Information Institute of Korea. The highest elevation is
350 m above sea level. The outcropping lithology of Halmidang Mountain consists of biotite gneiss (Bgn) and Pre-Cambrian
Era banded biotite gneiss (PCEbngn) (Geological Survey of Korea, 1972; Korea Institute of Energy and Resources, 1982).
Halmidang Mountain is almost surrounded by urban development such as residential districts, small-sized factories, and a golf
course.

Several landslide events resulted from heavy rain with a cumulative rainfall of approximately 400 mm over 24 hours (from
17:00 on July 26, 2011, to 17:00 on July 27, 2011) over the area of Yongin-si, causing damage (as shown in Figure 1(b)). A
total of 36 catastrophic shallow landslides occurred at Halmidang Mountain because of the heavy rain. Locations of initiation
sites of these shallow landsides and recordings of rainfall intensity and cumulative rainfall during the two days (July 26–27)
are shown in Figure 2 and Figure 3, respectively. Debris flows occurred along 21 watersheds between 13:00 and 17:00 on July
27, 2011, that were transformed from shallow slope failures, resulting in property damage (i.e., agricultural lands such as
paddies and fields and several small-sized factories were washed away by water and debris).

Opo station is the nearest weather station to Halmidang Mountain (as shown in Figure 2) and belongs to the automatic warning
system (AWS) operated by the Korea Meteorological Administration (KMA). Precipitation of more than 1,000 mm (i.e., more
than 50% of the yearly rainfall in 2011) was measured at the Opo station over one month (July 2011). A maximum rainfall
intensity of 64.5 mm/h occurred from 13:00 to 14:00 on July 27 (see Figure 3). Continuous rainfall of 221 mm occurred during
the five hours (11:00–16:00 on July 27) immediately before the landslide events when rainfall intensities were strong, which
are greater than 40 mm/h. In particular, cumulative rainfall over the two hours before the landslide events reached 113 mm
(i.e., 28% of the total rainfall during the two days, July 26 and 27, 2011).

## 3 Topographic data and landslide inventories

GIS-based topographic information is essential to evaluate slope stability on a regional scale. Considering the channelization
caused by the use of small grids (<10 m cell size), disregarding significant areas for landslide assessments (Park et al., 2016),
we utilized the digital elevation model (DEM) with a cell size of 10 m. This DEM was modeled using 1:5,000 digital maps



provided by the National Geographic Information Institute of Korea (see Figure 1). Considering a slope angle at each cell, infiltration and slope stability analyses were independently conducted.

Landslide inventories comprise information in terms of slope failures and are also important sources of data used to compare the information with the locations of potential slope failure areas predicted by slope stability assessments for performance evaluation. We built a total of 36 slope failure initiation sites in the GIS format (as shown in Figure 2) by comparing satellite images with a 5-m resolution of the area of Halmidang Mountain which were taken before and after the landslide events in 2011. We also checked the accuracy of the landslide inventories by comparing some of them with actual slope failure sites during our field investigations.

## 4 Methods

### 4.1 Infiltration in unsaturated conditions

We simulated infiltration of rainfall into the ground on unsaturated soil slopes by applying both the two-phase flow of water and air and the coupled hydro-mechanical model in FLAC 2D Ver. 7.0 (Itasca, 2011) program. Two immiscible fluids (i.e., wetting and non-wetting fluids; the wetting fluid moistens porous media (soil particles) more than the non-wetting fluid) can 100 be simulated to flow in a porous medium in FLAC using the two-phase flow option. These two fluids are assumed to occupy entire pore spaces of the porous media. Therefore, the sum of saturations of the wetting fluid ($S_w$) and the non-wetting fluid ($S_g$) can be regarded as 1 (i.e., $S_w + S_g = 1$, where subscripts, $w$ and $g$, denote wetting and non-wetting fluids, respectively). Water and air can be used to replace the wetting and non-wetting fluids, respectively, in unsaturated soils.

A relationship between effective saturation ($S_e$) and capillary pressure ($P_C$) (van Genuchten, 1980) is expressed by Eq. (1). The 105 capillary pressure is calculated by subtracting the wetting fluid pressure ($P_w$) from the non-wetting fluid pressure ($P_g$) (i.e., $P_C = P_g - P_w$).

$$S_e = \left[(P_C/P_0)^{\frac{1}{1-a}} + 1\right]^{-a} = \frac{S_w - S_w^r}{1 - S_w^r} \tag{1}$$

where $P_0$ is based approximately on air-entry value, $a$ is equal to the $m$ constant used for van Genuchten soil-water retention curve (SWRC), $S_w$ is the saturation of the wetting fluid, and $S_w^r$ is the residual saturation.

The matric suction ($\psi$) can be derived from Eq. (1) and can be expressed by Eq. (2).

$$\psi = P_C = P_0 \left[(S_e)^{-\frac{1}{a}} - 1\right]^{1-a} \tag{2}$$

The two-phase flow (wetting and non-wetting fluids) is calculated by Darcy's law. The fluid flow velocities ($q_i^w$ and $q_i^g$ for wetting and non-wetting fluids, respectively) are expressed by Eqs. (3) and (4).





$$q_i^w = -k_{ij}^w k_r^w \frac{\partial}{\partial x_j}(P_w - \rho_w g_k x_k) \tag{3}$$

$$q_i^g = -k_{ij}^w \frac{\mu_w}{\mu_g} k_r^g \frac{\partial}{\partial x_j}(P_g - \rho_g g_k x_k) \tag{4}$$

where $k_{ij}^w$ is the saturated mobility coefficient, $k_r^w$ and $k_r^g$ are the relative permeabilities of wetting and non-wetting fluids, respectively, $\frac{\mu_w}{\mu_g}$ is the viscosity ratio of the wetting fluid to the non-wetting fluid, $\rho_w$ and $\rho_g$ are the densities of the wetting and non-wetting fluids, respectively, and $g$ is the gravitational acceleration.

The relative permeability of a wetting fluid is used for calculating the unsaturated hydraulic conductivity. The relative
permeability of a wetting fluid ($k_r^w$) (van Genuchten, 1980) is generally used in FLAC (Eq. (5)). The relative permeability of a non-wetting fluid ($k_r^g$) (Lenhard and Parker, 1987) is applied in this study, expressed by Eq. (6).

$$k_r^w = (S_e)^b \{1 - [1 - (S_e)^{\frac{1}{a}}]^a\}^2 \tag{5}$$

$$k_r^g = (1 - S_e)^c [1 - (S_e)^{\frac{1}{a}}]^{2a} \tag{6}$$

where $b$ and $c$ are constants, both which are equal to 0.5 (i.e., Cho, 2016).
The mobility coefficient ($k$) depends on a relationship among the hydraulic conductivity ($k_s$), gravitational acceleration ($g$), and water density ($\rho_w$), as shown in Eq. (7).

$$k = k_s/(g\rho_w) \tag{7}$$

Balance equations for the wetting and non-wetting fluids (Eqs. (8) and (9)) can be derived from the fluid balance laws and the fluid constitutive laws. The balance of momentum is associated with the total stress ($\sigma$), bulk density ($\rho$), gravitational
acceleration ($g$), and velocity ($\dot{u}$), expressed by Eq. (10). The bulk density for unsaturated conditions is expressed as follows by Eq. (11).

$$n\left[\frac{S_w}{K_w}\frac{\partial P_w}{\partial t} + \frac{\partial S_w}{\partial t}\right] = -\left[\frac{\partial q_i^w}{\partial x_i} + S_w\frac{\partial \varepsilon}{\partial t}\right] \tag{8}$$

$$n\left[\frac{S_g}{K_g}\frac{\partial P_g}{\partial t} + \frac{\partial S_g}{\partial t}\right] = -\left[\frac{\partial q_i^g}{\partial x_i} + S_g\frac{\partial \varepsilon}{\partial t}\right] \tag{9}$$

where $n$ is the porosity, $K$ is the fluid bulk modulus, $\varepsilon$ is the volumetric strain, and $t$ is the time.

$$\frac{\partial \sigma_{ij}}{\partial x_j} + \rho g_i = \rho \frac{d\dot{u}_i}{dt} \tag{10}$$

- 5 -


$$\rho = \rho_d + n(S_w \rho_w + S_g \rho_g) \tag{11}$$

where $\rho_d$ is the density of the soil under dry conditions, $\rho_w$ is the density of a wetting fluid (water), and $\rho_g$ is the density of a non-wetting fluid (air).

The effective stress ($\sigma'_b$) given by Bishop (1959), expressed by Eq. (12), is utilized in the Mohr-Coulomb constitutive model
(Eq. (13)) to interpret the failure of soil. The matric suction coefficient ($\chi$) in Bishop's effective stress equation can be substituted by the saturation of a wetting fluid ($S_w$) (Vanapalli et al., 1996), and Eq. (12) can be organized into Eq. (14) considering that the sum of saturations of wetting and non-wetting fluids is equal to 1 (i.e., $S_w + S_g = 1$). Equation (13) is then organized into Eq. (15)

$$\sigma'_b = (\sigma - P_g) + \chi(P_g - P_w) \tag{12}$$

$$\tau_{\max} = \sigma'_b \tan\phi' + c' \tag{13}$$

$$\sigma'_b = \sigma - (S_w P_w + S_g P_g) \tag{14}$$

$$\tau_{\max} = [\sigma - (S_w P_w + S_g P_g)] \tan\phi' + c' \tag{15}$$

where $\tau_{\max}$ is the shear strength, $\phi'$ is the effective friction angle, $c'$ is the effective cohesion, and $S_w P_w + S_g P_g$ is the pore fluid pressure.

Deformation of microstructure in soil slopes is attributed to changes in stress and degree of saturation during rainfall infiltration, causing a change in porosity (Wu and Selvadurai, 2016). Such variations in void of soil cause a change in permeability (Hu et al., 2011). Chapuis and Aubertin (2003) used Kozeny–Carman equation (Eq. (16)) to predict the hydraulic conductivity of soil, which depends on a change in porosity. Because changes in hydraulic conductivity of soil during rainfall infiltration are not considered in FLAC, we programmed Eq. (16) to be applied during infiltration analysis using an in-built programming
language (FISH).

$$k_s = k_{s0} \left(\frac{n}{n_0}\right)^3 \left(\frac{1-n_0}{1-n}\right)^2 \tag{16}$$

**4.2 Slope stability analysis**

In general, rainfall-induced shallow slope failure in Korea is parallel to the ground surface. Therefore, the infinite slope model can possibly be applied for rainfall-induced shallow slope failure analysis (Cho, 2014). We applied the infinite slope model
and the limit equilibrium method to each cell of the DEM in the study area. The safety factor of the infinite slope ($F_s$) (Eq.





(17)) was derived by dividing the shear strength of the soil ($\tau_f$) by the shear stress on the failure surface ($\tau_m$) by applying the Mohr-Coulomb failure criterion (Eq. (15)).

$$F_s = \frac{\tau_f}{\tau_m} = \frac{[W\cos^2\beta - (S_w P_w + S_g P_g)]\tan\phi' + c'}{W \sin\beta \cos\beta} \qquad (17)$$

where $\beta$ is the angle of the infinite slope, and $W$ is the weight of a soil slice.

The weight of soil increases with the degree of saturation during rainfall. The weight of a soil slice on a potential failure surface is given by integrating the sum of the dry unit weight of soil ($\gamma_d$) and the unit weight of water occupying void spaces of partially saturated soil (Eq. (18)).

$$W = \int_0^{z_w} \gamma_t \, dz = \int_0^{z_w} [\gamma_d + n S_w(z)\gamma_w] \, dz \qquad (18)$$

where $z_w$ is the depth to the potential failure surface, $\gamma_t$ is the total unit weight of soil, $S_w(z)$ is the saturation which depends
on depth, and $\gamma_w$ is the unit weight of water.

## 5 Numerical study

### 5.1 Geotechnical and hydrological properties

We conducted detailed geotechnical field investigations at Halmidang Mountain by sampling soils at 37 sites (as shown in Figure 4) and conducting laboratory and field tests, such as direct shear test, soil-water characteristic test, field density test,
and field permeability test. Table 1 summarizes the representative soil properties (total/dry unit weight of soil, saturated hydraulic conductivity, cohesion, and internal friction angle) and soil-water characteristics (saturated and residual volumetric water contents) obtained from the laboratory and field tests. Hysteresis of soil-water characteristics was not considered in this study. We grouped sampling points where soil properties were similar and divided Halmidang Mountain into twelve zones (i.e., Zones 1 to 12) considering the distribution of watersheds where each group is located (see Figure 4). Table 2 summarizes the
representative soil properties and soil-water characteristics for each zone, which were determined as the average values at sampling points located in each zone. Because the SWRC coefficients (van Genuchten, 1980) are used in FLAC by default to consider soil-water characteristics, we applied van Genuchten SWRCs to Zones 1 to 12, as shown in Figure 5(a) (i.e., coefficients of van Genuchten SWRC at each zone are given in Table 2). Using such SWRC coefficients, we computed Mualem-van Genuchten relative permeability curves (van Genuchten, 1980) corresponding to Zones 1 to 12, as shown in
Figure 5(b). Table 3 summarizes the material properties of water and air required for simulations of rainfall infiltration. We assumed water to be incompressible.



**5.2 Slope geometry and initial and boundary conditions**

The depths from ground surfaces to slope failure surfaces are generally shallow with a range of 1–3 m in Korea (Kim et al., 2004). We assumed that a soil depth was uniform at 2 m and that the ground was isotropic and homogeneous. We modeled

infinite slopes considering a slope angle at each cell of the GIS-based topography of the study area and the material properties of soil given in Table 2. We also set displacements at the bottom surface of the infinite slopes to be fixed into both horizontal and vertical directions and set displacements at the left and right sides to be fixed into the horizontal direction. We assumed that the initial pore air pressure on the slopes was equivalent to the atmospheric pressure and that other boundaries were considered to be airtight in the two-phase flow option. In contrast, we set the pore air pressure to be zero in the single-phase

flow option. We applied the hourly rainfall (as shown in Figure 3) as water flux at the slope surface. Seoul Metropolitan Government (2014) reported that matric suction near-surface grounds in mountainous areas in central Korea was maintained at a range of 10–20 kPa before rainfall during the rainfall season. We assumed that slopes were steady-state conditions with a constant initial pore water pressure of -20 kPa by depth.

**5.3 Analysis results**

**5.3.1 Infiltration analysis**

We simulated rainfall infiltration at infinite slopes for a period of 22 hours (from 17:00 on 26 July to 15:00 on 27 July) by applying both the single-phase flow model and the coupled hydro-mechanical model which considers two-phase flow. Figure 6 presents an example of changes in pore water/air pressures and matric suction against time at Zones 1 to 12 when the coupled hydro-mechanical model was applied (i.e., the angle of the infinite slope was 30°, similar to the average slope angle at slope

failure sites with a value of 27°). Changes in pore water pressure and matric suction when the single-phase flow model was applied are also shown in Figure 7.

Increases in the pore water pressure at Zones 2, 4, 5, 10, and 11 when both models were applied were markedly faster than those at the other Zones because of a high infiltration capacity at Zones 2, 4, 5, 10, and 11 with large relative permeabilities (as shown in Figure 5b), although saturated hydraulic conductivities at entire Zones are similar except for a relatively high

value at Zone 2 (i.e., $8.87 \times 10^{-5}$ m/s). Before 20 hours from the start (17:00 on 26 July), ponding effects did not occur because of relatively small rainfall intensities compared to saturated hydraulic conductivities (see Figure 3 and Table 2). However, a slight air pressure was generated during rainfall infiltration, resulting in increases in pore water pressure when the coupled hydro-mechanical model was applied. Sequentially, increases in pore water pressure in the coupled hydro-mechanical model became slightly greater than those in the single-phase flow model. Pore water pressures from the start to 22 hours at Zones 2,

10, and 11 in the coupled hydro-mechanical model slightly increased quickly compared to those in the single-phase flow model without ponding effects because of the high infiltration capacity. In contrast, infiltration rates at the other Zones from the single-phase flow model became slightly faster than those in the coupled hydro-mechanical model after 20–22 hours. Matric



suction at Zones 1, 3, 4, 8, and 9 from both models became similar. In particular, matric suction at Zones 5, 6, 7, and 12 from the single-phase model became larger than that from the coupled hydro-mechanical model because water flow in the void

spaces was interrupted by air pressure during ponding (e.g., Dong et al., 2017; Wu et al., 2017; Liu et al., 2018) when the coupled hydro-mechanical model was applied. Air pressure significantly affects behavior of rainfall infiltration irrespective of ponding occurring.

Although porosity decreases as deformation progresses during rainfall infiltration into the slope, resulting in the reduction of saturated hydraulic conductivity (Hu et al., 2011), pore water pressure in the coupled hydro-mechanical model increased more

quickly than that in the single-phase flow model before ponding. Increases in pore water pressure caused by air pressure possibly had greater effects on the behavior of rainfall infiltration compared with the reduction of the increase rate of pore water pressure caused by decreases in porosity during deformation because the settlement rate of slope is slow at the early stage of infiltration but sharply increases when the ground becomes saturated (Dong et al., 2017). Slower infiltration rates in the coupled hydro-mechanical model after an occurrence of ponding, compared with that in the single-phase flow model, were

possibly affected by the reduction of porosity as well as the interruption by air pressure.

Wetting bands at Zones 1, 3, 6, 7, 8, 9, and 12 slowly progressed into the depths and did not reach the bottom of soil layers because of a low infiltration capacity with small relative permeability. Matric suction at those Zones in the coupled hydro-mechanical model was small at shallow depths but did not dissipate (i.e., pore air pressure is slightly greater than positive pore water pressure at shallow depths, as shown in Figure 6). Therefore, the upper regions did not become fully saturated. In contrast,

matric suction in the single-phase flow model gradually dissipated from the surface without the generation of pore air pressure (see Figure 7).

Infiltration rate was the fastest at Zone 10 because of the largest relative permeabilities near an initial matric suction of 20 kPa, and the ground was completely saturated after 16 hours. The pore water pressure at Zone 2 with the largest saturated hydraulic conductivity ($8.87 \times 10^{-5}$ m/s) slowly increased compared to that at Zones 4, 10, and 11 because of the small relative

permeabilities at a matric suction of less than 20 kPa (see Figure 3). The saturated hydraulic conductivity at Zone 5 is the smallest, but increases in the pore water pressure were faster than those at Zones 1, 3, 6, 7, 8, 9, and 12 because of large relative permeabilities. Therefore, the effect of relative permeability is more sensitive on infiltration rates than that of saturated hydraulic conductivity.

**5.3.2 Slope stability assessment**

We obtained landslide susceptibility maps for Halmidang Mountain based on GIS-based topographic data (i.e., DEM and slope map with a cell size of 10 m). The geotechnical properties of Zones 1 to 12 are given in Table 2, and Eq. (17) with the distributions of pore water and air pressures at 0, 8, 16, 22 hours (Figure 8 and Figure 9). The distribution patterns for the safety factor from both models from the start to 16 hours are similar because the distributions of matric suction from both





models until 16 hours barely differ without ponding effects under rainfall intensities smaller than the infiltration capacity (see

Figure 6 and Figure 7). Although infiltration rates after 22 hours from both models became different because of heavy rainfall, distributions patterns of the safety factor at Zones 1, 3, 7, 8, and 12 from both models are barely different with no area of predicted slope failure (i.e., safety factor<1) because decreases in safety factor at those Zones are too small due to relatively large shear strength parameters and slow infiltration rates with small relative permeabilities. This result can be compared with no actual slope failure site existing in those Zones. In contrast, matric suctions at Zones 4 and 10 from both models dissipated

after 22 hours because of fast infiltration rates with large relative permeabilities, resulting in distribution patterns for safety factor at Zones 4 and 10 from both models to be almost the same. Thus, there is no significant difference in the distribution patterns of the safety factor from both models where infiltration progress is too slow to reduce safety factor or when the ground is saturated, although infiltration rates from both models are different.

Final matric suctions at Zones 2 and 11 from the coupled hydro-mechanical model were smaller than those from the single-

phase flow model after 22 hours, as shown in Figure 6 and Figure 7. Therefore, safety factors at Zones 2 and 11 from the coupled hydro-mechanical model were smaller than those from the single-phase flow model, and areas of predicted slope failure in Zones 2 and 11 from the coupled hydro-mechanical model are more widely distributed than those from the single-phase flow model (i.e., the predicted slope failure areas in Zone 2 from both models are 33,000 and 18,900 $m^2$, respectively, and those in Zone 11 from both models are 559,500 and 407,000 $m^2$, respectively). In contrast, the final matric suctions at

Zones 5, 6, and 9 from the single-phase model were smaller than those from the coupled hydro-mechanical model (see Figure 6 and Figure 7). Therefore, safety factors at Zone 5 from the single-phase flow model were smaller than those from the coupled hydro-mechanical model, and the area of predicted slope failure in Zone 5 from the single-phase flow model was more widely distributed than those from the coupled hydro-mechanical model (i.e., the predicted slope failure areas from both models are 353,700 and 229,200 $m^2$, respectively). However, safety factors at Zones 6 and 9 from the single-phase flow model were

slightly larger than those from the coupled hydro-mechanical model. This is because minimum safety factors from the coupled hydro-mechanical model occurred at a depth of 1.2 m because of the decrease in the effective stress from the large positive pore water pressure induced by air pressure at the upper part of the slope when ponding occurred (as shown in Figure 6 and Figure 7), which became smaller than minimum safety factors from the single-phase flow model. The total area of potential slope failure in the coupled hydro-mechanical model (i.e., 1,350,500 $m^2$) is slightly larger than that of the single-phase flow

model (i.e., 1,307,500 $m^2$).

The relative permeabilities at Zones 2, 4, 5, 10, and 11 are larger than those at the other Zones (as shown in Figure 5). Therefore, the predicted slope failure areas are widely distributed in those Zones, whereas the potential slope failure areas are markedly small in the other Zones. This is comparable with the fact that actual slope failure sites were mainly distributed in Zones 2, 4, 5, 10, and 11.





Figure 10 shows changes in the safety factor from the coupled hydro-mechanical and the single-phase flow models against time (0, 8, 16, and 22 hours) at 36 actual slope failure sites, among which 1, 4, 14, 1, 7, and 9 actual slope failure sites were respectively located in Zones 2, ,4, 5, 9, 10, and 11. The initial safety factor for the actual slope failure sites is the largest at Zone 11 with a value of 2.76 because of the gentle slope gradient (18.1°) and large internal friction angle (30.7°) and is the smallest at Zone 5 with a value of 1.22 because of the stiff slope gradient (31.8°) and the small cohesion (4.3 kPa) and internal

friction angle (24.4°). Safety factors gradually decreased from the initial values at Zone 10 and reduced to less than 1 the fastest (after 16 hours) because of the most rapid infiltration, as shown in Figure 10(e). Safety factors slightly decreased at Zones 2, 4, 5, and 11 and sharply reduced after 20 hours, as shown in Figure 10(a), (b), (c), and (f). Many safety factors at those Zones (24/28) reduced to less than 1 after 22 hours. In contrast, the final safety factors remained larger than 1.5 at Zone 9 after 22 hours due to slow infiltration rates, as shown in Figure 10(d). Although initial safety factors at Zone 10 are larger than those

at Zone 2, the safety factors at Zone 10 rapidly reduced to less than 1. Therefore, hydrological properties (i.e., saturated hydraulic conductivity and relative permeability) affecting infiltration rates significantly affect changes in the safety factor.

Safety factors from the coupled hydro-mechanical and single-phase flow models were similar during the early stages because of weak rainfall, but the decreases in the safety factor from the coupled hydro-mechanical model became greater than those of the single-phase flow model during the middle stages because of fast increases in pore water pressure affected by air pressure.

Increases in both pore water and air pressures from the coupled hydro-mechanical model resulted in a reduction in shear strength (see Eq. (17)). Contrastingly, the decrease in shear strength from the single-phase flow model was only caused by the increase in pore water pressure. The safety factor during the last stage shows various trends depending on properties of infiltration at the zones. At Zones 2, 9, and 11, safety factors from the single-phase flow model remained greater than those from the coupled hydro-mechanical model. At Zones 4 and 10, safety factors from both models became equivalent as the

grounds were saturated. At Zone 5, the safety factors from the single-phase flow model after 22 hours became smaller than those from the coupled hydro-mechanical model because the grounds in the single-phase flow model became saturated earlier than those in the coupled hydro-mechanical model, as shown in Figure 6 and Figure 7. Changes in the safety factor are sensitive to rates of rainfall infiltration depending on whether the actual infiltration mechanism is considered by applying both air flow and hydro-mechanical coupling as well as hydrological properties of the ground. Therefore, the proper model needs to be

selected to precisely conduct infiltration and slope stability analyses.

We then applied the modified success rate (MSR) (Huang and Kao, 2006) to assess the quantitative performance for prediction of slope failure areas. Equation (19) defines MSR which can consider the performance for prediction of slope failure areas in both unstable and stable areas. MSR has the advantage of avoiding under- and over-prediction (with a range of 0–1). Huang and Kao (2006) reported that MSR can derive the best simulation when the stable cell coverage over a total watershed area is

in a range of 80–90%. Landslide under-prediction is likely to occur when the stable cell coverage is larger than 90%, and over-prediction is likely when the stable cell coverage is less than 80%.





$$\text{MSR} = 0.5 \times \text{SR} + 0.5 \times \frac{\text{successfully predicted stable cells}}{\text{total number of actual stable cells}} \quad\quad (19)$$

where SR is the former success rate ($= \frac{\text{number of successfully predicted slope failures}}{\text{total number of actual slope failures}}$).

Among the 36 actual slope failures in Halmidang Mountain, 32 and 31 slope failures were successfully predicted by the coupled
hydro-mechanical and single-phase flow models, respectively. Therefore, the SRs for both models are 0.89 (32/36) and 0.86 (31/36), respectively. MSRs for both models are 0.9 and 0.89, respectively, and the stable cell coverages over the total watershed area are 90.7% and 91%, which slightly exceed the range for the best simulation (i.e., 80–90%). Both models showed good performance for prediction of slope failure areas. The difference is not significant for this case because ponding barely occurred because of large saturated hydraulic conductivities of soils in the study area and a short duration of heavy rain.
However, considering that the coupled hydro-mechanical model results in considerable differences in safety factor compared to the uncoupled model when the rainfall similar to a saturated hydraulic conductivity is consistently applied (e.g., Wu et al., 2016; Wu et al., 2017), properly applying hydro-mechanical coupling to regions where ponding is likely to occur with a low infiltration capacity is necessary. The SR and MSR from the coupled hydro-mechanical model are slightly larger than those from the single-phase flow model. Given the performance values produced by the MSR and SR, the coupled hydro-mechanical
model possesses the best agreement with the actual landslide events at Halmidang Mountain.

## 6. Conclusions

Although air flow and hydro-mechanical coupling significantly influence the infiltration behavior of rainfall and changes in the safety factor of unsaturated soils, it has barely been considered at regional and local scales. We applied the fully coupled hydro-mechanical model considering two-phase (water and air) flow to simulate rainfall infiltration and slope stability and
evaluated the applicability of the model for the Halmidang Mountain where landside events occurred in July 2011. We used the available GIS-based topographic data, geotechnical and hydrological properties, and historical rainfall data to build the model. We interpreted changes in the safety factor at the actual slope failure sites and the MSR index to assess the performance of the model. Observations and insights from results of the infiltration and slope stability analyses are as follows:

1) Air flow and hydro-mechanical coupling consistently influence changes in pore water pressure on unsaturated slopes during
rainfall infiltration. The slight air pressure generated before ponding occurs under weak rainfall results in slightly larger increases in pore water pressure than those from the single-phase flow model, although porosity decreases as deformation progresses, resulting in a slight reduction of saturated hydraulic conductivity. Air flow and reduction of saturated hydraulic conductivity caused by deformation interrupt rainfall infiltration after ponding occurs under heavy rain, causing infiltration rates to be slower than those from the single-phase flow model. Therefore, applying the coupled hydro-mechanical model to
accurately simulate rainfall infiltration considering the actual infiltration mechanism is necessary.



2) The safety factor prior to rainfall is mainly affected by shear strength parameters (cohesion and internal friction angle). However, infiltration rate strongly affects changes in the safety factor during rainfall infiltration. Increases in pore water pressure affected by air pressure in the coupled hydro-mechanical model become larger than those in the single-phase flow model before ponding occurs during weak rain, and hence, safety factors decrease rapidly compared with those in the single-

phase flow model. However, slopes in the single-phase flow model are rapidly saturated after ponding occurs during heavy rain because of the quick rates of rainfall infiltration without the interruption by air flow, and hence, final safety factors become equivalent or smaller than 1 before those in the coupled hydro-mechanical model do.

3) Increases in pore water pressure depending on the model type have significant effects on landslide susceptibility maps. MSR gives a better performance value for the coupled hydro-mechanical model compared with the single-phase flow model.

Therefore, infiltration and slope stability analyses using the coupled hydro-mechanical model have a good applicability to evaluate landslide events in Halmidang Mountain. The coupled hydro-mechanical model is expected to modify the accuracy of simulating landslide events and to contribute to the prevention of damage caused by landslides in other regions with the validation of the applicability of this model for other landslide cases.

**Data availability**

The digital maps used in this study are available at: http://map.ngii.go.kr/ms/map/NlipMap.do. The precipitation data used in this study are available at: http://www.weather.go.kr/weather/climate/past_cal.jsp. The field-collected data are included in Tables 1 and 2 of this paper.

**Author contribution**

All authors significantly contributed to the research. Sinhang Kang conducted analyses and wrote this manuscript. Byungmin

Kim designed this research.

**Competing interests**

The authors declare that they have no conflict of interest.



**Acknowledgments**

This research was supported by the National Research Foundation of Korea (NRF) funded by the Ministry of education (NRF-
2018R1A6A3A01012888) and by the grant (19SCIP-B146946-02) from Construction technology research program funded by
Ministry of Land, Infrastructure and Transport of Korean government.

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





# TABLE

**Table 1: Representative soil properties and soil-water characteristics obtained from laboratory and field tests.**

| Sample no. | Coordinate of sampling location | | $\gamma_t$ (kg/m$^3$) | $\gamma_d$ (kg/m$^3$) | $k_s$ (m/s) | $c$ (kPa) | $\phi$ (°) | Volumetric water content | |
|---|---|---|---|---|---|---|---|---|---|
| | N | E | | | | | | $\theta_s$ | $\theta_r$ |
| 1 | 37°19'22.3" | 127°12'20.3" | 1400 | 1169 | $5.76\times10^{-5}$ | 5.72 | 30.52 | 0.40 | 0.15 |
| 2 | 37°19'00.5" | 127°11'56.0" | 1770 | 1440 | $3.59\times10^{-5}$ | 7.82 | 34.32 | 0.53 | 0.32 |
| 3 | 37°19'13.9" | 127°12'39.8" | 1860 | 1584 | $2.5\times10^{-5}$ | 15.22 | 32.72 | 0.44 | 0.19 |
| 4 | 37°19'32.4" | 127°12'27.9" | 1860 | 1351 | $4.13\times10^{-5}$ | 8.50 | 30.20 | 0.49 | 0.29 |
| 5 | 37°19'41.8" | 127°11'37.7" | 1790 | 1536 | $3.07\times10^{-5}$ | 7.12 | 28.70 | 0.36 | 0.15 |
| 6 | 37°19'41.8" | 127°11'24.1" | 1530 | 1286 | $4.71\times10^{-5}$ | 14.91 | 33.71 | 0.45 | 0.16 |
| 7 | 37°19'54.7" | 127°12'14.9" | 1770 | 1423 | $3.64\times10^{-5}$ | 4.90 | 32.10 | 0.46 | 0.23 |
| 8 | 37°19'45.9" | 127°13'02.8" | 1790 | 1515 | $3.03\times10^{-5}$ | 8.04 | 33.89 | 0.41 | 0.18 |
| 9 | 37°19'10.4" | 127°13'25.6" | 1670 | 1445 | $3.79\times10^{-5}$ | 34.42 | 30.63 | 0.40 | 0.22 |
| 10 | 37°19'52.3" | 127°12'39.8" | 1815 | 1581 | $2.71\times10^{-5}$ | 52.66 | 28.07 | 0.34 | 0.15 |
| … | … | … | … | | | … | … | … | … |

$\gamma_t$: Total unit weight of soil; $\gamma_d$: Dry unit weight of soil; $k_s$: Saturated hydraulic conductivity; $c$: Cohesion; $\phi$: Internal friction
angle; $\theta_s$: Saturated volumetric water content; and $\theta_r$: Residual volumetric water content.





**Table 2: Summary of representative soil properties and soil-water characteristics for each zone used in the infiltration and slope failure analyses, which were determined as the average values at sampling points located in each zone.**

| Zone no. | $\gamma_t$ (kN/m³) | $\gamma_d$ (kN/m³) | $k_s$ (m/s) | $c$ (kPa) | $\phi$ (°) | Volumetric water content | | van Genuchten SWRC coefficient | | |
|---|---|---|---|---|---|---|---|---|---|---|
| | | | | | | $\theta_s$ | $\theta_r$ | $\alpha$ | n | m |
| 1 | 16.4 | 13.4 | $4.74\times10^{-5}$ | 15.8 | 33.3 | 0.37 | 0.18 | 0.833 | 1.29 | 0.22 |
| 2 | 16.8 | 14.4 | $8.87\times10^{-5}$ | 10.1 | 27.3 | 0.4 | 0.17 | 0.137 | 1.5 | 0.33 |
| 3 | 15.3 | 12.8 | $4.9\times10^{-5}$ | 18 | 31.4 | 0.39 | 0.18 | 0.769 | 1.33 | 0.25 |
| 4 | 17.8 | 14.8 | $4.14\times10^{-5}$ | 6.4 | 26 | 0.4 | 0.22 | 0.103 | 1.5 | 0.33 |
| 5 | 18 | 15.2 | $3.02\times10^{-5}$ | 4.3 | 24.4 | 0.35 | 0.19 | 0.143 | 1.4 | 0.29 |
| 6 | 16.5 | 13.7 | $3.64\times10^{-5}$ | 5.2 | 32.1 | 0.38 | 0.19 | 0.556 | 1.29 | 0.22 |
| 7 | 17.9 | 15.4 | $3.07\times10^{-5}$ | 7.1 | 28.7 | 0.36 | 0.15 | 0.526 | 1.32 | 0.24 |
| 8 | 15.3 | 12.9 | $4.89\times10^{-5}$ | 9.4 | 34.5 | 0.43 | 0.17 | 0.633 | 1.32 | 0.24 |
| 9 | 15.9 | 13 | $4.68\times10^{-5}$ | 6.8 | 32 | 0.47 | 0.23 | 0.556 | 1.3 | 0.23 |
| 10 | 18.4 | 15.1 | $4.13\times10^{-5}$ | 8 | 26.8 | 0.42 | 0.12 | 0.056 | 1.3 | 0.23 |
| 11 | 17.3 | 14.6 | $3.57\times10^{-5}$ | 2.6 | 30.7 | 0.35 | 0.14 | 0.107 | 1.54 | 0.35 |
| 12 | 16.8 | 14.5 | $3.72\times10^{-5}$ | 11 | 21 | 0.35 | 0.14 | 0.357 | 1.26 | 0.21 |

$\gamma_t$: Total unit weight of soil; $\gamma_d$: Dry unit weight of soil; $k_s$: Saturated hydraulic conductivity; $c$: Cohesion; $\phi$: Internal friction angle; $\theta_s$: Saturated volumetric water content; and $\theta_r$: Residual volumetric water content.

**Table 3: Material properties of air and water (Cho, 2016) applied to the rainfall infiltration analysis.**

| Parameter | Value |
|---|---|
| Viscosity ratio, $\mu_w/\mu_a$ | 56 |
| Air density, $\rho_a$ | 1.25 kg/m³ |
| Water density, $\rho_w$ | 1000 kg/m³ |
| Bulk modulus of air, $K_a$ | $1\times10^5$ Pa |
| Bulk modulus of water, $K_w$ | $2\times10^9$ Pa |


# FIGURES

(a)

(b)

**Figure 1: (a) Location and elevation distributions of Halmidang Mountain. Sources of basemap: Esri, DigitalGlobe, GeoEye, Earthstar Geographics, CNES/Airbus, USDA, USGS, AeroGRID, IGN, and the GIS User Community. (b) Damage caused by landslides events that occurred in Yongin-si in July 2011.**






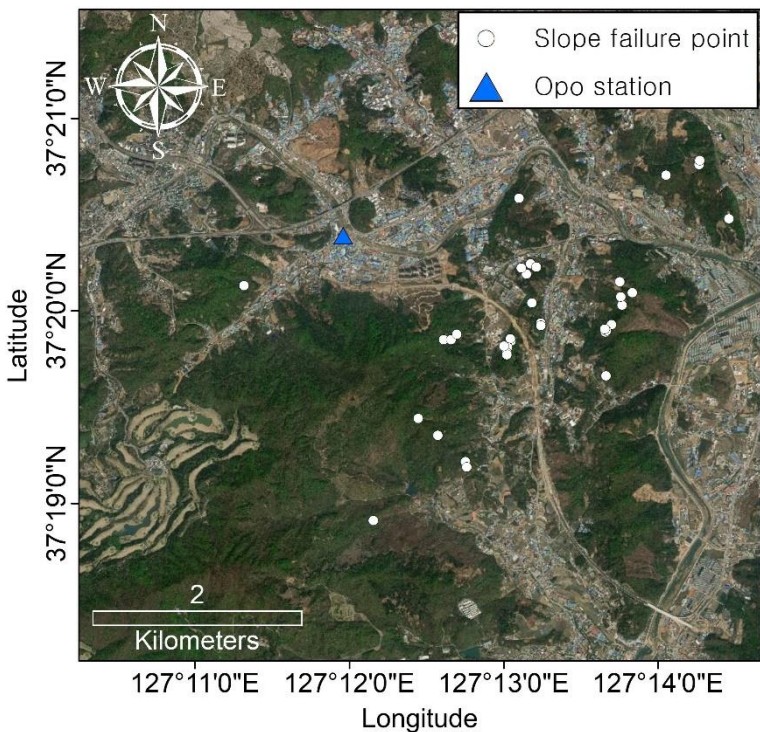

Figure 2: Locations where slope failure occurred in the area of Halmidang Mountain in 2011 and location of the Opo station. Sources of basemap: Esri, DigitalGlobe, GeoEye, Earthstar Geographics, CNES/Airbus, USDA, USGS, AeroGRID, IGN, and the GIS User Community.





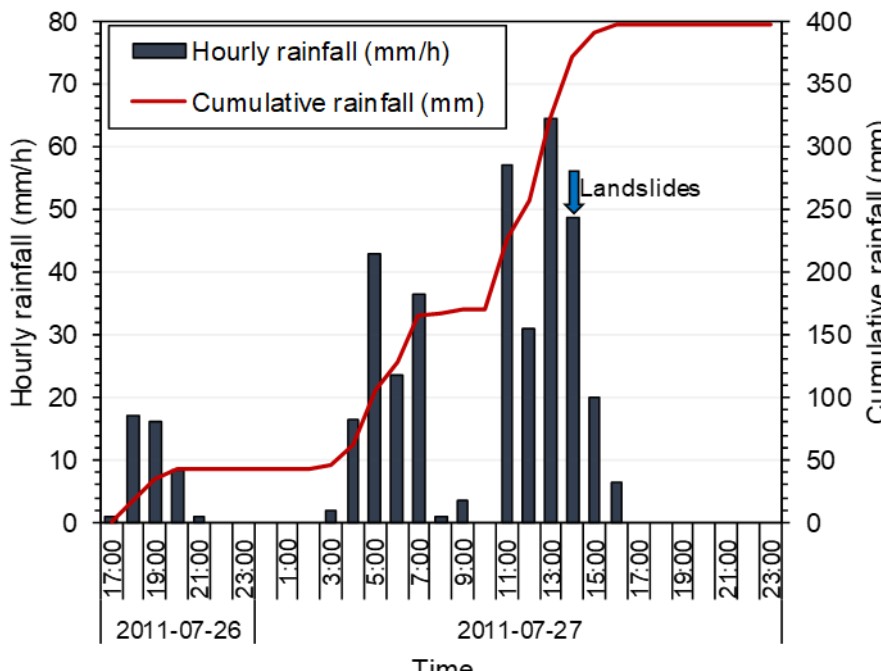

**Figure 3: Hourly and cumulative rainfall recorded at the Opo station during the two days from July 26 to 27, 2011 (obtained from KMA).**


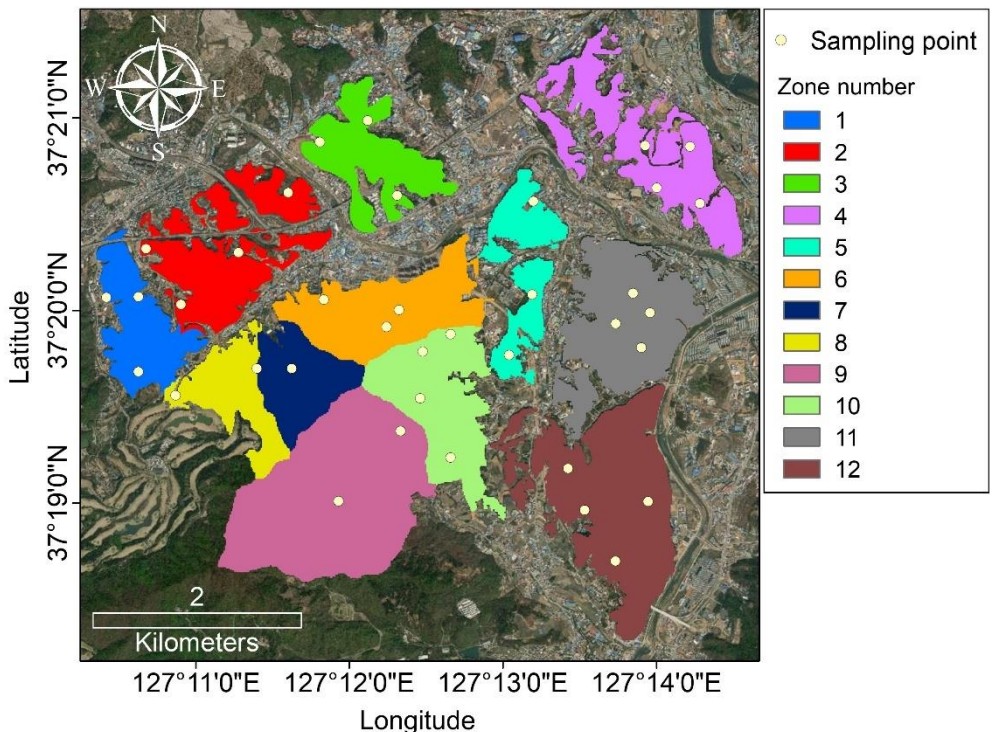

Figure 4: Locations where soil sampling was conducted to obtain geotechnical properties and the distribution of the twelve zones. Sources of basemap: Esri, DigitalGlobe, GeoEye, Earthstar Geographics, CNES/Airbus, USDA, USGS, AeroGRID, IGN, and the GIS User Community.


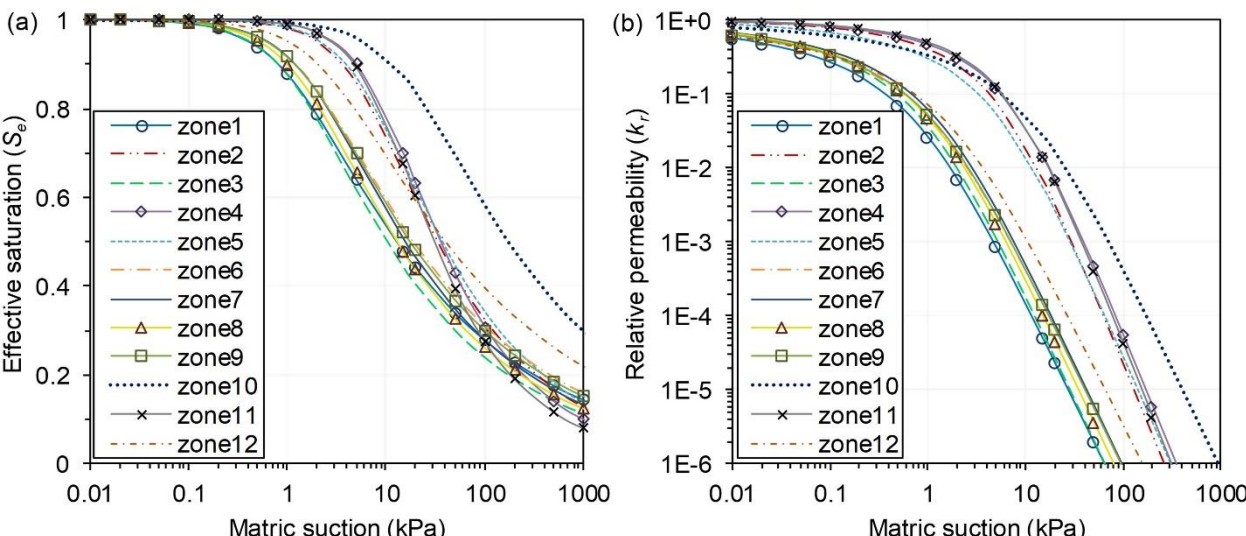

**Figure 5: (a) van Genuchten soil-water retention curves (SWRCs) for Zones 1 to 12. (b) Relative permeability ($K_r$) curves computed from the twelve SWRCs shown in Figure 5(a). van Genuchten SWRC constants used to compute Mualem-van Genuchten relative permeability curves are given in Table 2.**



**Figure 6: Changes in pore water/air pressures and matric suction against time (from 0 to 22 h) when the coupled hydro-mechanical**
**model was applied to an infinite slope model with an angle of 30°. The vertical axes of graphs represent the depth from the slope surface. The columns represent ranges of pore water/air pressures and matric suction, respectively.**





**Figure 6: continued**





**Figure 6: continued**



**Figure 7: Changes in pore water pressure and matric suction against time (from 0 to 22 h) when the single-phase flow model was applied to an infinite slope model with an angle of 30°. The vertical axes of graphs represent the depth from the slope surface. The columns represent ranges of pore water pressure and matric suction, respectively.**







**Figure 7: continued**

- 28 -




**Figure 7: continued**


**Figure 8: Locations of slope failures and changes in landslide susceptibility map, presenting distributions of safety factors against time (0, 8, 16, and 22 hours) from the coupled hydro-mechanical model and slope stability assessment. Sources of basemap: Esri, DigitalGlobe, GeoEye, Earthstar Geographics, CNES/Airbus, USDA, USGS, AeroGRID, IGN, and the GIS User Community.**


**Figure 9: Locations of slope failures and changes in landslide susceptibility map, presenting distributions of safety factors against** 520 **time (0, 8, 16, and 22 hours) from the single-phase flow model and slope stability assessment. Sources of basemap: Esri, DigitalGlobe, GeoEye, Earthstar Geographics, CNES/Airbus, USDA, USGS, AeroGRID, IGN, and the GIS User Community.**




**Figure 10:** Changes in safety factor at 36 actual slope failure sites against time: (a) 1 site in Zone 2; (b) 4 sites in Zone 4; (c) 14 sites in Zone 5; (d) 1 site in Zone 9; (e) 7 sites in Zone 10; and (f) 9 sites in Zone 11. Similar color and same symbol are used at a line expressing changes in safety factor at a site. To distinguish results from the coupled hydro-mechanical and the single-phase flow models, solid and dashed lines are used, respectively.