# Peer review of "Effects of coupled hydro-mechanical model considering two-phase fluid flow on potential for shallow landslides: a case study in Halmidang Mountain, Yongin, South Korea"

_Natural Hazards and Earth System Sciences, 2019_

## Referee Comment (RC1) · Anonymous Referee #1 · 6 Dec 2019

The paper studies the effect of two phase fluid flow analysed in a coupled hydro-mechanical manner on rainfall infiltration modelling and slope stability at a regional scale. The research also looks into the variation of hydraulic conductivity through the Kozeny-Carman equation owing to the soil deformation under subsurface water infiltration. The study is interesting in that the authors seek to extend the two-phase coupled hydro-mechanical behaviour under rainfall infiltration from the previously slope-scale limited study to a regional scale and also, strives to understand the changes in hydraulic conductivity owing to infiltration-induced deformation. However, the research

lacks novelty and has significant shortcomings with regard to the methodology and fails to impress upon the reader the need for such a complex undertaking instead of the traditional single-phase modelling at a regional scale.

Highlighted below are the major issues in the study:

1) The use of the Kozeny-Carman equation (16) to link the volume changes in unsaturated soil with the variation of saturated hydraulic conductivity (ks) doesn't seem reasonable. The Kozeny-Carman equation is used to roughly predict the vertical saturated hydraulic conductivity for homogenised soils. As far as I am aware Chapuis and Aubertin (2003) or any other studies have not tested the equation to model volume change behaviours like swelling or collapse under saturated or unsaturated conditions. Whilst procedures to measure the volume change in unsaturated condition exists, the quantification of corresponding changes in hydraulic conductivity owing to the volume changes under rainfall infiltration is a task yet to be accomplished. The paper also doesn't explain how the Kozeny-Carman equation for saturated hydraulic conductivity is used to model effective stress changes and the subsequent variations in unsaturated hydraulic conductivity under rainfall infiltration. In light of the above shortcoming, the reviewer thinks the review of the results presented in the paper as of now would be a fruitless exercise.

2) The authors have not clearly explained how the two-dimensional model for seepage analysis (FLAC) has been applied at a regional scale. Has the subsurface-flow routing from different grid cells been considered? Also, any attempts at validation cannot be seen (e.g. field-based monitoring, streamflow data from gauge stations, etc.).

3) Did the authors use the effective stress estimated during the hydro-mechanical coupled seepage analysis in FLAC in the assessment of the factor of safety? Please provide a detailed explanation in section 4.2.

4) It is difficult to follow the motivation of the authors in conducting the two-phase coupled hydro-mechanical based infiltration modelling at a regional scale. No information

could be found in the paper with regard to the volume change behaviour of soils from Central Korea under unsaturated conditions (soil volume changes under wetting and drying). This is a fundamental issue the authors need to sort out before attempting to model at any scale. Also, under circumstances of volume change, authors would require to carry out SWCC corrections for volume change as well.

5) Another drawback is the lack of description with regard to the field mapped landslide characteristics, evidence from sites in all zones with regard to soil profiles (single or several different layers) and soil depth especially when field investigations were carried out (mentioned in Section 3). Please provide necessary details.

6) Vanapalli et al. (1996) used two approaches to calculate the shear strength. The first approach was to use a dimensionless number "normalised area of water with k as a fitting parameter" and the second approach was to use a normalised degree of saturation (defined as effective saturation in this paper) wherein the residual degree of saturation needs to be estimated. The authors in this study have substituted Bishop's matric suction coefficient with the saturation of a wetting fluid variable (Equation 12 and Equation 14). Could the authors explain the basis for equating the degree of saturation (instead of an effective saturation) of a wetting fluid with the Bishop's matric suction coefficient?

7) The authors have focused more on the modelling aspect with advanced two-phase modelling at the regional scale and did not worry much about the variability in input data which clearly will influence the safety factor values. It is recommended that such a study (sensitivity analysis) be undertaken in the region. Also, could the authors explain why only watershed criteria was used in creating the zones? Why wasn't geological information used? Please explain in detail.

8) The reasoning for the selection of a 10-m DEM is not clear (Section 3). Why is channelisation important for slope stability analysis? I can understand its importance in debris flow modelling. Please explain by also including information with regard to the

size of landslides mapped.

---

## Author Comment (AC1) · 12 Feb 2020

**Effects of coupled hydro-mechanical model considering two-phase fluid flow on potential for shallow landslides: a case study in Halmidang Mountain, Yongin, South Korea**

**Sinhang Kang and Byungmin Kim** *

We thank the referee for the insightful comments which truly helped enrich the manuscript. In the revised manuscript, we have clarified contributions of the manuscript. We have also applied the changed coupled hydro-mechanical model considering deformation-dependent water retention behavior with hydraulic hysteresis. For the comments raised by the reviewers, we have provided the point by point responses.

| Referee #1's Comments | Responses |
|---|---|
| The research lacks novelty and has significant shortcomings with regard to the methodology and fails to impress upon the reader the need for such a complex undertaking instead of the traditional single-phase modelling at a regional scale. | We revised the following sentences in Lines 49–64 to complement contributions of this work. "*Because air flow delays wetting process on soil slope associated with rainfall infiltration (Hu et al., 2011), a neglect of air flow would result in an imprecise simulation (Laloui et al., 2003), such as an overestimation of deformation induced by rainfall infiltration (Hu et al., 2016). Effects of deformation on water retention behavior should be considered in the collapse during wetting process (Hu et al., 2016). Water retention curve hysteresis is fundamental for the soil–water–air coupling (Ebel et al., 2010; Tsai, 2011; Borja et al., 2012; Yang et al., 2017), and it has significant effects on distribution of water content and slope stability (Ma et al., 2011). Whereas it has been demonstrated that the coupled hydro-mechanical model considering two-phase fluid flow and deformation-dependence of water retention behavior with hydraulic hysteresis accurately simulates the behavior of unsaturated deformable soils at a slope scale (e.g., Hu et al.,* |

| | *2016; Hu et al., 2018), such models have rarely been applied to evaluate slope stability on a regional scale.* |
|---|---|
| | *Considering efficient uses of computing resources, we simplified slopes at cells of the GIS-based topography of Halmidang Mountain located in Yongin-si, South Korea to be infinite slopes in a two-dimensional domain. We applied the coupled hydro-mechanical model based on numerical methods to those infinite slopes for suitable simulations of slope failure induced by rainfall infiltration. The changes in pore air/water pressures and void ratios obtained from the simulation of rainfall-infiltration were used as input data for slope failure analyses at each infinite slope model, and the minimum safety factor on the infinite slope was determined to be a safety factor of the corresponding cell of the GIS-based topography."* |
| The use of the Kozeny-Carman equation (16) to link the volume changes in unsaturated soil with the variation of saturated hydraulic conductivity ($k_s$) doesn't seem reasonable. The Kozeny-Carman equation is used to roughly predict the vertical saturated hydraulic conductivity for homogenised soils. As far as I am aware Chapuis and Aubertin (2003) or any other studies have not tested the equation to model volume change behaviours like swelling or collapse under saturated or unsaturated conditions. Whilst procedures to measure the volume change in unsaturated condition exists, the quantification of corresponding changes in hydraulic conductivity owing to the volume changes under rainfall | Several previous studies have used Kozeny–Carman equation incorporated in coupled hydro-mechanical models to compute the saturated permeability varied depending on porosity (e.g., Chapuis and Aubertin, 2003; Cho, 2016a; Kim et al., 2016; Kim et al., 2018). Changes in hydraulic conductivity owing to volume changes under rainfall infiltration have not exactly quantified, but it is clear that variations in void of soil affect permeability (Hu et al., 2011), and Hu et al. (2013) and Hu et al. (2018) have applied an equation to predict the permeability of deformable unsaturated soils. We also applied their equation incorporated in the coupled hydro-mechanical model. We revised the following sentences in Lines 190–198.

 *"Chapuis and Aubertin (2003), Cho (2016a), Kim et al. (2016), and Kim et al. (2018) have used Kozeny–Carman equation incorporated in coupled hydro-mechanical models to compute the saturated* |

infiltration is a task yet to be accomplished. The paper also doesn't explain how the Kozeny-Carman equation for saturated hydraulic conductivity is used to model effective stress changes and the subsequent variations in unsaturated hydraulic conductivity under rainfall infiltration. In light of the above shortcoming, the reviewer thinks the review of the results presented in the paper as of now would be a fruitless exercise.

*permeability which depends on porosity. Hu et al. (2013) and Hu et al. (2018) have applied Eq. (20) to predict the permeability of deformable unsaturated soils, which depends on changes in porosity and void ratio. Because changes in permeability of the soil deformed during rainfall infiltration are not considered in FLAC, we programmed Eq. (20) to be applied during infiltration analysis using an in-built programming language (FISH).*

$$k(e) = \frac{k_0}{n_0{}^2 exp(2k_p e_0)} n^2 exp(2k_p e) \qquad (20)$$

*where $k_0$ is the initial permeability, $n_0$ is the initial porosity, $k_p$ is the parameter involved in Eq. (3), and $e_0$ is the initial void ratio."*

We added the following sentences in Lines 199–205 to describe the scheme of a coupled hydro-mechanical model and how Eq. (20) was incorporated in the model.

*"The coupled hydro-mechanical model consists of the fluid flow, mechanical, and water retention model loops. The fluid flow loop evaluates fluid flows from pressure gradients and changes in saturation and pore pressure due to unbalanced flows, based on from Eq. (7) to Eq. (13). The mechanical loop evaluates total stress depending on velocities, coordinates, and generation of pore pressure due to mechanical volume strain, based on from Eq. (14) to Eq. (19). The water retention model loop updates saturation and permeability that depend on mechanical volumetric change and generation of pore pressure, based on from Eq. (3) to Eq. (6) and Eq. (20). The stress state sequentially updated from the modified water retention behavior is applied to the next time step for the fluid flow loop."*

The authors have not clearly explained how the two-dimensional model for

We added the following sentences in Lines 207–214 in section 4.1 to describe how to apply a coupled

| | |
|---|---|
| seepage analysis (FLAC) has been applied at a regional scale. Has the subsurface-flow routing from different grid cells been considered? Also, any attempts at validation cannot be seen (e.g. field-based monitoring, streamflow data from gauge stations, etc.). | hydro-mechanical model for infiltration analyses at a regional scale.

 "*We applied the coupled hydro-mechanical model for simulations of rainfall infiltration to the independent 2D infinite slope model considering slope angles from different cells of the slope raster computed from the DEM in the study area. The depths from ground surfaces to slope failure surfaces observed during our field investigation are generally shallow and comparable with a range of 1–3 m associated with Korea (Kim et al., 2004). We set a uniform soil depth and a length of an infinite slope to be 2 m and 10 m, respectively, and applied the soil properties obtained from field investigations. Finally, saturations and pore pressures of wetting and non-wetting fluids (water and air) could be computed at all area of the infinite slope for a period of 22 hours from starting the simulations.*"

 We added the section 5.1 (Lines 232–263) for validation of the coupled hydro-mechanical model using the experimental results obtained from Liakopoulos (1964). |
| Did the authors use the effective stress estimated during the hydro-mechanical coupled seepage analysis in FLAC in the assessment of the factor of safety? Please provide a detailed explanation in section 4.2. | We added the following sentences in Lines 227–230 in section 4.2 to describe how to determine safety factors.

 "*We evaluated slope stability of infinite slope models for a period of 22 hours based on Eq. (20) utilizing the variations in saturations and pore pressures of water and air with time simulated from the coupled hydro-mechanical model. The minimum safety factors of infinite slope models were finally determined to be safety factors of different cells of the GIS-based topography of the study area.*" |
| It is difficult to follow the motivation of the authors in conducting the two-phase coupled hydro-mechanical based | We revised the sentences in Lines 49–64 to complement contributions of this work, as shown in the response to the first comment. |

infiltration modelling at a regional scale. No information could be found in the paper with regard to the volume change behaviour of soils from Central Korea under unsaturated conditions (soil volume changes under wetting and drying). This is a fundamental issue the authors need to sort out before attempting to model at any scale. Also, under circumstances of volume change, authors would require to carry out SWCC corrections for volume change as well.

Volume change of unsaturated soils depends on changes in matric suction and net normal stress (Matyas and Radhakrishna, 1968; Fredlund and Rahardjo, 1993). Soils in Korea would also be expected to follow this relationship among them. We added the following sentence in Lines 39–43 to describe why the volume change behavior of unsaturated soils should be used.

"*Considering that volume of unsaturated soils changes depending on matric suction and net normal stress (Matyas and Radhakrishna, 1968) and relationship among them can be considered to make the constitutive equation for volumetric strain of unsaturated soils (Fredlund and Rahardjo, 1993), a coupled hydro-mechanical model considering the volume change behavior can be applied to simulate hydraulic processes in unsaturated soils.*"

We applied the deformation-dependent water retention curve model (Hu et al., 2013; Hu et al., 2016) to consider both volume change of unsaturated soils and water retention curve hysteresis, as shown in the following sentences in Lines 127–146, and we corrected SWRC model parameters in Table 3.

"*The hydraulic hysteresis reflects different hydraulic states and hydraulic paths, and the saturation of the wetting fluid for deformable soils depends on the soil skeleton deformation as well as matric suction (Hu et al., 2013; Hu et al., 2016). The water retention behavior is classified into two groups, the main wetting and drying surfaces and the scanning curves. Eq. (1) can be replaced by Eq. (3), which defines a bounding surface (wetting or drying) considering the hysteretic water retention behavior for deformable soils subjected to mechanical and hydraulic loading.*

$$S_{e,\gamma}(\psi, e) = \left[\{\beta_\gamma exp(k_p e)\psi\}^{\frac{1}{1-a}} + 1\right]^{-a}, \ \gamma = w, d \qquad (3)$$

*where $\beta_\gamma$ is the air entry value (for main drying surface, $\beta_\gamma = \beta_d$, and for main wetting surface, $\beta_\gamma = \beta_w$), $k_p$ is the model parameter, and $e$ is the void ratio.*

*Hu et al. (2013) considered the incremental effective saturation associated with scanning zones during movement of a soil state, expressed by Eq. (4). Integrating Eq. (4) from $S_{e,n}$ to $S_{e,n+1}$, $\psi_n$ to $\psi_{n+1}$ $(= \psi_n + d\psi_n)$, and $e_n$ to $e_{n+1}$ $(= e_n + de_n)$, Eq. (6) can be obtained to compute the updated trial effective saturation $(S_{e,n+1}^{trial})$.*

$$dS_e = -S_e\left(1 - S_e^{-1/a}\right)\left(k_{ss}\frac{d\psi}{\psi} + k_{se}de\right) \qquad (4)$$

$$\begin{cases} \frac{\partial ln S_e}{\partial ln\psi} = -k_{ss}\left(1 - S_e^{1/a}\right) \\ \frac{\partial ln S_e}{\partial e} = -k_{se}\left(1 - S_e^{1/a}\right) \end{cases} \qquad (5)$$

$$S_{e,n+1}^{trial} = \left[1 - \frac{(\psi_{n+1})^{k_{ss}/a}exp\left(\frac{k_{se}}{a}e_{n+1}\right)}{(\psi_n)^{k_{ss}/a}exp\left(\frac{k_{se}}{a}e_n\right)}\left(1 - S_{e,n}^{-1/a}\right)\right]^{-a} \qquad (6)$$

*where $k_{ss}$ and $k_{se}$ are the slopes of the asymptotes for the scanning curves in the $ln S_e - ln\psi$ and $ln S_e - e$ planes, respectively.*

*The following procedure is required to determine the updated saturation $(S_{e,n+1})$:*

*If $S_{e,n+1}^{trial} < S_{e,d}(\psi_{n+1}, e_{n+1})$ and $S_{e,n+1}^{trial} > S_{e,w}(\psi_{n+1}, e_{n+1})$ then $S_{e,n+1} \leftarrow S_{e,n+1}^{trial}$; else if $S_{e,n+1}^{trial} \geq S_{e,d}(\psi_{n+1}, e_{n+1})$ then $S_{e,n+1} \leftarrow S_{e,d}(\psi_{n+1}, e_{n+1})$; else $S_{e,n+1}^{trial} \leq S_{e,w}(\psi_{n+1}, e_{n+1})$ then $S_{e,n+1} \leftarrow S_{e,w}(\psi_{n+1}, e_{n+1})$.*"

| | |
|---|---|
| Another drawback is the lack of description with regard to the field mapped landslide characteristics, evidence from sites in all zones with regard to soil profiles (single or several different layers) and soil depth especially when field investigations were carried out (mentioned in Section 3). Please provide necessary details. | We added the following sentences in Lines 104–109 in Section 3 to supply additional information about landslides in the study area and observations during field investigations. "*From the slope failures, a total of 21 debris flows were transformed with a total debris flow spreading area of approximately 94,000 m². Areas and distances of debris flow spreading ranged from 1,100 to 19,600 m² and from 90 to 580 m,* |

| | |
|---|---|
| | *respectively. We checked the accuracy of the landslide inventories by comparing some of them with actual slope failure sites during our field investigations. Figure 2(b) shows the slope failure initiation sites we observed. Failure surfaces were within depths to weathered rocks up to which soils consisted of a single layer. Depths from ground surfaces to slope failure surfaces were generally shallow within a range from 1.3 to 2.1 m."* |
| Vanapalli et al. (1996) used two approaches to calculate the shear strength. The first approach was to use a dimensionless number "normalised area of water with k as a fitting parameter" and the second approach was to use a normalised degree of saturation (defined as effective saturation in this paper) wherein the residual degree of saturation needs to be estimated. The authors in this study have substituted Bishop's matric suction coefficient with the saturation of a wetting fluid variable (Equation 12 and Equation 14). Could the authors explain the basis for equating the degree of saturation (instead of an effective saturation) of a wetting fluid with the Bishop's matric suction coefficient? | Some previous studies have used degree of saturation to be the matric suction coefficient ($\chi$) in Bishop's effective stress equation (e.g., Chateau and Dormieux, 2001, 2002; Cho, 2016b; Hu et al., 2018; Zhang et al., 2018). We revised the following sentence in Lines 176–179.

 *"The matric suction coefficient ($\chi$) in Bishop's effective stress equation can be substituted by the saturation of a wetting fluid ($S_w$) (Chateau and Dormieux, 2001, 2002; Cho, 2016b; Hu et al., 2018; Zhang et al., 2018), and Pham et al. (2019) reported that critical points computed from effective stress utilizing the $S_w$ were close to saturated critical state line with large correlations statistically evaluated."* |
| The authors have focused more on the modelling aspect with advanced two-phase modelling at the regional scale and did not worry much about the variability in input data which clearly will influence the safety factor values. It is recommended that such a study (sensitivity analysis) be undertaken in the region. Also, could the authors explain why only watershed criteria was | We added Figure 13 and the following sentences in Lines 416–430 to describe results of the sensitivity analysis.

 *"Limited number of samples were used to determine representative material properties of the study area in spite of complex geological features and variability in material properties. We investigated effects of cohesion ($c$), saturated hydraulic conductivity ($k_s$), water retention model parameter ($k_p$), and van Genuchten SWRC coefficient ($a$) on* |

used in creating the zones? Why wasn't geological information used? Please explain in detail.

*characteristics of change in safety factor. Figure 13 shows variations in safety factor with time at an infinite slope model with an angle of 30° when material properties of Zone 10 were consistently applied with the exception of changing only $c$ or $k_s$ or $k_p$ or $a$. As a value of cohesion became large from 0 to 9 kPa, an initial safety factor increased from 1.4 to 1.95 (Figure 13(a)). The rates of decrease in safety factor were not affected by cohesion. It is observed in Figure 13(b) that safety factors slowly and continuously decreased when saturated hydraulic conductivity was small ( $k_s = 3 \times 10^{-5}$ m/s). However, the greater the saturated hydraulic conductivity, the larger the reduction in safety factor when rainfall occurred (from 0 to 5 h and from 12 to 22 h), and the smaller the reduction in safety factor when rainfall did not occur (from 6 to 11 h). When the water retention model parameter decreases, an air entry pressure ($P_0$) becomes large, and a rate of increase in degree of saturation with a decrease in matric suction becomes fast. Therefore, the smaller the water retention model parameter, the faster the reduction in safety factor (Figure 13(c)). As a van Genuchten SWRC coefficient increases, the slope gradient of water retention curve becomes steep, and a degree of saturation at the same matric suction becomes small. A large SWRC coefficient that results in slow rates of increase in degree of saturation affects the reduction in safety factor to be slow (Figure 13(d)).*"

As described in the "Study area" section, the study area consists of same geological system (biotite gneiss). Thus, we used only the watershed to classify zones.

| The reasoning for the selection of a 10-m DEM is not clear (Section 3). Why is channelisation important for slope stability analysis? I can understand its importance in debris flow modelling. Please explain by also including information with regard to the size of landslides mapped. | We corrected the following sentence in Lines 95–98 to simplify descriptions about why we applied a 10-m DEM.

 "*Considering the cell size of digital elevation model (DEM) used in previous studies which have evaluated physically based models for predicting landslides at a regional scale (e.g., Park et al., 2016; Salvatici et al., 2018; Park et al., 2019), we utilized the DEM with a cell size of 10 m.*"

 We added the following sentences in Lines 104–109 to supply additional information about landslides in the study area.
 "*From the slope failures, a total of 21 debris flows were transformed with a total debris flow spreading area of approximately 94,000 m². Areas and distances of debris flow spreading ranged from 1,100 to 19,600 m² and from 90 to 580 m, respectively. We checked the accuracy of the landslide inventories by comparing some of them with actual slope failure sites during our field investigations. Figure 2(b) shows the slope failure initiation sites we observed. Failure surfaces were within depths to weathered rocks up to which soils consisted of a single layer. Depths from ground surfaces to slope failure surfaces were generally shallow within a range from 1.3 to 2.1 m.*" |
|---|---|

---

## Referee Comment (RC2) · Anonymous Referee #2 · 18 Apr 2020

General comments

After reading the discussion version, the first referee's comment and author's reply to that comment of this article, I post this comment to this article. I totally agree to the first referee's comment with eight questions which are truly considerable.

The research focusing on the regional shallow landslide susceptibility emphasizes on the mechanism based on the coupled hydro-mechanical model. The novelty in this paper is to consider two-phase fluid flow of water and air in the regional area, aiming

at obtaining the changes in pore and water pressures and saturations of air and water. However, the research lacks the detailed information and the full evidences explaining the reason why the coupled hydro-mechanical model is better the single-phase flow model.

Specific comments

Applying the physical mechanical is not easy, more information should be added, and the research work should be described.

1. In line 7, compared to the detailed information in section of study area, "More than 30 shallow landslides" is not clear. Please revise it.

2. The accuracy comparison. In line 17, your result with coupled hydro-mechanical model is "slightly" more consistent with the single-phase flow model. Although your presentation in the whole paper is good, the result is just slightly better. Also in Line 314 and line 317, the accuracy comparison of 0.89 vs 0.86 and 90.7% vs 91%. What is the meaning of your research?

3. In line 65, the paper mentioned the outcropping lithology. This area only includes two kinds of lithology, or this two lithology are the main types? Detailed lithology information should be described, better with a map, if necessary.

4. In line 71, 36 shallow landslide occurred at Halmidang Mountain. In line 73, debris flow occurred along 21 watersheds. In line 89 landslide inventories comprise information. In line 90, you applied performance evaluation. In line 93, you checked the accuracy of the landslide inventories. Please add the landslide inventory in this manuscript. What are types of these 36 natural hazard? As I cannot understand the meaning of landslide in your manuscript. The landslide means the natural hazards in the broad concept, or specific debris flow. Also please simply describe the work of performance evaluation. Please describe the accuracy of the landslide inventories.

5. In line 200-206, I cannot understand how to make the figure6 and figure7? To be

specific, in the zone 1 of figure 6, there are eleven points in the time line of 0h. Could you please describe it?

6. In the line 204, why you set the infinite slope 30°? The case study is a regional area. Are the conditions in the 12 zones the same?

7. In section 5.3.1, you aim to compare the coupled hydro-mechanical and single phase flow model. Please do not neglect the parameter sensitivity. For example, in figure 6, the plot of line and point are very similar in zone 1, zone 3 and zone 8. I see the parameters in Table 2, the parameters are not similar. Please explain.

8. What is the criterion of the division of 12 zones? As you divide the whole area into 12 zones, then the number of zone should be added into the Table I.

9. In the figure 2, the landslide occurs at 14:00. All or several the landslides happened at that time? Please support detailed information.

Technical corrections

1. Table I, please check the unit and the value of $\gamma t$ and $\gamma d$. The detailed information of all samples should be added.

2. Table II, please define the $\alpha$, n and m.

3. Please zoom in two panels in Figure 5.

---

## Author Comment (AC2) · 29 May 2020

**Effects of coupled hydro-mechanical model considering two-phase fluid flow on potential for shallow landslides: a case study in Halmidang Mountain, Yongin, South Korea**

**Sinhang Kang and Byungmin Kim** [*]

We thank the referee for their insightful comments which truly helped enrich the manuscript. In the revised manuscript, we have clarified contributions of the manuscript. We have also added detailed descriptions of the methodology. For the comments raised by the reviewer, we have provided the point by point responses.

| Referee #2's Comments | Responses |
|---|---|
| The research lacks the detailed information and the full evidences explaining the reason why the coupled hydro-mechanical model is better the single-phase flow model. | Numerous previous studies have proved that the coupled hydro-mechanical model could simulate infiltration behavior more appropriately than the single-phase flow model (e.g., Hu et al., 2011, 2016, 2018; Wu and Selvadurai, 2016). We added the following sentences explaining usefulness of the coupled model compared to the single-phase model in Line 49–55. *"Because air flow delays wetting process on soil slope associated with rainfall infiltration (Hu et al., 2011), a neglect of air flow would result in an imprecise simulation (Laloui et al., 2003), such as an overestimation of deformation induced by rainfall infiltration (Hu et al., 2016). Effects of deformation on water retention behavior should be considered in the collapse during wetting process (Hu et al., 2016). Water retention curve hysteresis is fundamental for the soil–water–air coupling (Ebel et al., 2010; Tsai, 2011; Borja et al., 2012; Yang et al., 2017), and it has significant effects on distribution of water content and slope stability (Ma et al., 2011)."* |

| | We also demonstrated that the coupled hydro-mechanical model simulated the experimental results obtained from Liakopoulos (1964) quite accurately in the section 5.1. Our purpose is to propose a simplified method applicable to the shallow landslide assessment at a regional scale utilizing the fully coupled hydro-mechanical model and to check the applicability of the method in forecasting shallow landslides at a regional scale. |
| | |
| | We added detailed descriptions of the coupled hydro-mechanical model composed of three loops (i.e., fluid flow, mechanical, and water retention model loops are sequentially applied to each time step) in the section 4.1. Descriptions of all the equations used in the coupled model and how to apply the model to 2-D infinite slopes are also given in the same section. We also added descriptions of how to conduct the slope stability analysis using results from the coupled model in the section 4.2. |
| In line 7, compared to the detailed information in section of study area, "More than 30 shallow landslides" is not clear. Please revise it. | We changed "More than 30 shallow landslides" to "36 shallow landslides" in Line 7. |
| The accuracy comparison. In line 17, your result with coupled hydro-mechanical model is "slightly" more consistent with the single-phase flow model. Although your presentation in the whole paper is good, the result is just slightly better. Also in Line 314 and line 317, the accuracy comparison of 0.89 vs 0.86 and 90.7% vs 91%. What is the meaning of your research? | Even though results from the coupled hydro-mechanical model is just slightly better than those from the single-phase flow model, it is necessary to apply improved methodology to simulate the actual phenomenon more accurately. While rainfall infiltrates into soils, both air existing in void spaces and changes in area of void spaces caused by soil deformation affect rainfall infiltration rates. We tried to propose a simplified method applicable to the shallow landslide assessment at a regional scale utilizing the fully coupled hydro-mechanical model and checked its usefulness in a forecast of slope failure applying it to actual landslide events in |

Korea. We revised the following sentences in Lines 49–66 to clearly describe the reason why we applied the coupled model and the contributions of this work.

*"Because air flow delays wetting process on soil slope associated with rainfall infiltration (Hu et al., 2011), a neglect of air flow would result in an imprecise simulation (Laloui et al., 2003), such as an overestimation of deformation induced by rainfall infiltration (Hu et al., 2016). Effects of deformation on water retention behavior should be considered in the collapse during wetting process (Hu et al., 2016). Water retention curve hysteresis is fundamental for the soil–water–air coupling (Ebel et al., 2010; Tsai, 2011; Borja et al., 2012; Yang et al., 2017), and it has significant effects on distribution of water content and slope stability (Ma et al., 2011). Whereas it has been demonstrated that the coupled hydro-mechanical model considering two-phase fluid flow and deformation-dependence of water retention behavior with hydraulic hysteresis accurately simulates the behavior of unsaturated deformable soils at a slope scale (e.g., Hu et al., 2016; Hu et al., 2018), such models have rarely been applied to evaluate slope stability on a regional scale.*

*We proposed a simplified method applicable to the shallow landslide assessment at a regional scale utilizing the fully coupled hydro-mechanical model and checked its usefulness in a forecast of slope failure applying it to actual landslide events in Korea. Considering efficient uses of computing resources, we simplified slopes at cells of the GIS-based topography of Halmidang Mountain located in Yongin-si, South Korea to be infinite slopes in a two-dimensional domain. We applied the coupled hydro-mechanical model based on numerical*

| | |
|---|---|
| | *methods to those infinite slopes for suitable simulations of slope failure induced by rainfall infiltration. The changes in pore air/water pressures and void ratios obtained from the simulation of rainfall-infiltration were used as input data for slope failure analyses at each infinite slope model, and the minimum safety factor on the infinite slope was determined to be a safety factor of the corresponding cell of the GIS-based topography.*" |
| In line 65, the paper mentioned the outcropping lithology. This area only includes two kinds of lithology, or this two lithology are the main types? Detailed lithology information should be described, better with a map, if necessary. | The mountainous area in the study area only includes two kinds of lithology (biotite gneiss and Pre-Cambrian Era banded biotite gneiss). We added Figure 2(a) that shows geological map of the study area and revised the following sentence in Lines 78–81 to complement the lithology information. *"The outcropping lithology of Halmidang Mountain consists of biotite gneiss (Bgn) and Pre-Cambrian Era banded biotite gneiss (PCEbngn), and that of the area surrounding the mountain consists of quaternary alluvium (Qa) and Quartzofeldspathic gneiss (Qgn), as shown in Figure 2(a) (Geological Survey of Korea, 1972; Korea Institute of Energy and Resources, 1982)."* |
| In line 71, 36 shallow landslide occurred at Halmidang Mountain. In line 73, debris flow occurred along 21 watersheds. In line 89 landslide inventories comprise information. In line 90, you applied performance evaluation. In line 93, you checked the accuracy of the landslide inventories. Please add the landslide inventory in this manuscript. What are types of these 36 natural hazard? As I cannot understand the meaning of landslide in your manuscript. The landslide means the natural hazards in the broad concept, or specific debris | In this study, the inventory means the locations where shallow slope failures initiated. The inventory map composed of slope failure occurrence points is shown in Figure 2(b). We will provide it as an electronic supplement which can usefully be utilized in other studies. We stated debris flows, which were initiated from 29 out of 36 slope failure occurrence areas and spread along 21 watersheds, to describe overall landslide damage in the study area. For clarity, we revised the following sentences in Lines 108–110. *"From 29 out of 36 slope failure occurrence areas, debris flows were transformed and spread along 21 watersheds with a total debris flow spreading area* |

| flow. Also please simply describe the work of performance evaluation. Please describe the accuracy of the landslide inventories. | *of approximately 94,000 m².* *Spreading area and distance of each debris flow ranged from 1,100 to 19,600 m² and from 90 to 580 m, respectively."*

As we described in the following sentence in Lines 106–108, 36 natural hazards were used as slope failures in this study.
*"We built a total of 36 slope failure initiation sites in the GIS format (as shown in Figure 2(b)) by comparing satellite images with a 5-m resolution of the area of Halmidang Mountain which were taken before and after the landslide events in 2011."*

The performance evaluation of models for a shallow slope failure prediction is conducted by comparing the inventory map with the locations of slope failure prediction areas. We already described it in the following sentence in Lines 104–106.
*"Landslide inventories comprise information in terms of slope failures and are also important sources of data used to compare the information with the locations of potential slope failure areas predicted by slope stability assessments for performance evaluation."*

We checked the accuracy of the landslide inventories by visiting actual slope failure occurrence sites and comparing coordinates of the inventories with those measured at the site during our field investigations. We added Figure 2(c) showing two actual slope failure sites we observed. We revised the following sentence in Lines 109–110 to clearly describe how to check the accuracy.
*"We checked the accuracy of the landslide inventories by comparing coordinated of some of them with actual coordinates of slope failure sites that we measured during our field investigations."* |

| | |
|---|---|
| In line 200-206, I cannot understand how to make the figure6 and figure7? To be specific, in the zone 1 of figure 6, there are eleven points in the time line of 0h. Could you please describe it? | In Figure 8 and Figure 9, eleven points you mentioned display profiles of pore water/air pressure or matric suction at 0, 4, 8, 12, 16, 20, and 22 hours from starting rainfall at depths from ground surface (0 m) to 2 m with 0.2-m intervals at the middle of infinite slope. From results of infiltration analysis on the infinite slope, the profiles of pore water/air pressure or matric suction could be obtained.
*Figure 6 and Figure 7 were changed to Figure 8 and Figure 9, respectively, in a revised manuscript. |
| In the line 204, why you set the infinite slope 30°? The case study is a regional area. Are the conditions in the 12 zones the same? | Figure 8 and Figure 9 show just examples of infiltration behaviors at the infinite slope with an angle of 30°. As we stated in Lines 300–301, the angle of 30° was just considered because it is similar to the average slope angle at actual slope failure sites in the study area with a value of 27°. Actually, however, numerous infinite slopes were modeled depending on slope angles of different cells of the slope raster computed from the DEM in the study area, as described in sentences in Lines 214–216. |
| In section 5.3.1, you aim to compare the coupled hydro-mechanical and single phase flow model. Please do not neglect the parameter sensitivity. For example, in figure 6, the plot of line and point are very similar in zone 1, zone 3 and zone 8. I see the parameters in Table 2, the parameters are not similar. Please explain. | Geotechnical parameters of zone 1, zone 3, and zone 8 are not similar in Table 2, but an infiltration behavior is almost dependent on hydraulic properties, such as saturated hydraulic conductivity and relative permeability. Values of saturated hydraulic conductivity of zone 1, zone 3, and zone 8 are almost the same (i.e., $4.74\times10^{-5}$, $4.9\times10^{-5}$, and $4.89\times10^{-5}$ m/s). Relative permeability curves of zone 1, zone 3, and zone 8 are also similar, as shown in Figure 7. We also added Figure 13 and the following sentences in Lines 422–436 to describe results of the sensitivity analysis.
"*Limited number of samples were used to determine representative material properties of the study area in spite of complex geological features and variability in material properties. We investigated effects of cohesion ( c ), saturated hydraulic* |

| | |
|---|---|
| | *conductivity ($k_s$), water retention model parameter ($k_p$), and van Genuchten SWRC coefficient ($a$) on characteristics of change in safety factor. Figure 13 shows variations in safety factor with time at an infinite slope model with an angle of 30° when material properties of Zone 10 were consistently applied with the exception of changing only $c$ or $k_s$ or $k_p$ or $a$. As a value of cohesion became large from 0 to 9 kPa, an initial safety factor increased from 1.4 to 1.95 (Figure 13(a)). The rates of decrease in safety factor were not affected by cohesion. It is observed in Figure 13(b) that safety factors slowly and continuously decreased when saturated hydraulic conductivity was small ($k_s = 3 \times 10^{-5}$ m/s). However, the greater the saturated hydraulic conductivity, the larger the reduction in safety factor when rainfall occurred (from 0 to 5 h and from 12 to 22 h), and the smaller the reduction in safety factor when rainfall did not occur (from 6 to 11 h). When the water retention model parameter decreases, an air entry pressure ($P_0$) becomes large, and a rate of increase in degree of saturation with a decrease in matric suction becomes fast. Therefore, the smaller the water retention model parameter, the faster the reduction in safety factor (Figure 13(c)). As a van Genuchten SWRC coefficient increases, the slope gradient of water retention curve becomes steep, and a degree of saturation at the same matric suction becomes small. A large SWRC coefficient that results in slow rates of increase in degree of saturation affects the reduction in safety factor to be slow (Figure 13(d))."* |
| What is the criterion of the division of 12 zones? As you divide the whole area into 12 zones, then the number of zone should be added into the Table I. | First of all, we grouped sampling points where soil properties (i.e., unit weight, cohesion, internal friction angle, saturated hydraulic conductivity, and soil classification) were similar, and then the watersheds where sampling points belonging to the |

| | same group were combined to create a zone. As described in the "Study area" section, the study area consists of the same geological system (biotite gneiss). Thus, we used only the soil properties and watershed to classify zones. We revised the following sentences in Lines 274–276.
*"We grouped the 37 sampling points where soil properties were similar and divided Halmidang Mountain into twelve zones (i.e., Zones 1 through 12) based on watersheds to which the groups of sampling points belong (see Figure 6)."*
\*Figure 4 was changed to Figure 6 in a revised manuscript.

We added the number of zone in Table 2.
\*Table 1 was changed to Table 2 in a revised manuscript. |
|---|---|
| In the figure 3, the landslide occurs at 14:00. All or several the landslides happened at that time? Please support detailed information. | As we described in Lines 86–88, debris flows occurred along 21 watersheds between 13:00 and 15:00 on July 27, 2011, that were transformed from shallow slope failures. We added the following sentence in Line 88–89.
*"We assumed an occurrence time of slope failures to be 14:00 on July 27, 2011 to simplify analyses."* |
| Table I, please check the unit and the value of $\gamma_t$ and $\gamma_d$. The detailed information of all samples should be added. | We corrected values of $\gamma_t$ and $\gamma_d$ in Table 2. Considering that the information of all samples were not directly used for analyses in this study, we can provide it as a supplementary file. |
| Table II, please define the $\alpha$, n and m. | While we responded to reviews from Referee #1, we applied $k_{ss}$, $\beta_d$, $\beta_w$, $k_p$, and $a$ (Hu et al., 2013) for van Genuchten SWRC instead of $a$, $n$, and $m$. We added definitions of them in the footnote of Table 3.
\*Table 2 was changed to Table 3 in a revised manuscript. |
| Please zoom in two panels in Figure 5. | We corrected ranges of matric suction in two panels in Figure 7 from 0.01–1000 kPa to 0.1–100 kPa to zoom in the existing panels. |

| | |
|---|---|
| | *Figure 5 was changed to Figure 7 in a revised manuscript. |